# Activation of Nrf2/HO-1 by Peptide YD1 Attenuates Inflammatory Symptoms through Suppression of TLR4/MYyD88/NF-κB Signaling Cascade

**DOI:** 10.3390/ijms22105161

**Published:** 2021-05-13

**Authors:** Md Saifur Rahman, Md Badrul Alam, Young Kyun Kim, Mst Hur Madina, Ismail Fliss, Sang Han Lee, Jin Cheol Yoo

**Affiliations:** 1Department of Pharmacy, College of Pharmacy, Chosun University, Gwangju 501-759, Korea; md-saifur.rahman.1@ulaval.ca (M.S.R.); arirangkyk1114@naver.com (Y.K.K.); 2Department of Food Science, Faculty of Agriculture and Food Sciences, Laval University, Québec, QC G1V 0A6, Canada; ismail.fliss@fsaa.ulaval.ca; 3Department of Food Science and Biotechnology, Kyungpook National University, Daegu 702-701, Korea; mbalam@knu.ac.kr; 4Department of Phytology, Faculty of Agriculture and Food Sciences, Laval University, Québec, QC G1V 0A6, Canada; mosammad-hur.madina.1@ulaval.ca

**Keywords:** anti-inflammation, YD1, peptide drug, NF-κB, RAW 264.7, TLR-4, MyD88, AKT

## Abstract

In this study, we investigate the immunomodulatory effects of a novel antimicrobial peptide, YD1, isolated from Kimchi, in both in vitro and in vivo models. We establish that YD1 exerts its anti-inflammatory effects via up-regulation of the Nrf2 pathway, resulting in the production of HO-1, which suppresses activation of the NF-κB pathway, including the subsequent proinflammatory cytokines IL-1β, IL-6, and TNF-α. We also found that YD1 robustly suppresses nitric oxide (NO) and prostaglandin E2 (PGE_2_) production by down-regulating the expression of the upstream genes, iNOS and COX-2, acting as a strong antioxidant. Collectively, YD1 exhibits vigorous anti-inflammatory and antioxidant activity, presenting it as an interesting potential therapeutic agent.

## 1. Introduction

Low molecular weight bioactive peptides are increasingly recognized as strong candidates for therapeutic agents due to their broad spectrum of activity in the innate immune system. Microbial fermentation is extensively used for producing critical proteolytic enzymes, which can efficiently hydrolyze proteins to produce bioactive peptides [1]. We previously identified a glycine-rich low molecular weight peptide, YD1, which displays potent antimicrobial and antioxidant activity [2,3]. YD1 is purified from the probiotic *Bacillus amyloliquefaciens CBSYD1*, isolated from Korean traditional fermented Kimchi. However, the specific cellular and molecular anti-inflammatory mechanisms of YD1 have yet to be elucidated.

Inflammation is an intricate protective immune and physiological response to injured tissues. This response involves the activation of various immune cells, including monocytes and macrophages [4]. Then again, macrophage cells serve as the first line of defense in infected cells, and activated macrophages are a major source of ROS, and RNS triggers epigenetic changes, leading to the pathogenesis of the chronic disease. Thus, the activated macrophage models, RAW264.7, can identify active components for the development of a functional diet through a multiple target strategy, and in a laboratory setup, RAW 264.7 murine macrophage cells are suitable transfection hosts [5]. Chronic dysregulation of the inflammatory immune response can lead to the development of numerous severe disease states, including arthritis, cancer, heart disease, obesity, neurological disorders, and diabetes [6]. Therefore, targeting the dysfunctional inflammatory process is a feasible approach for therapeutic purposes. Pattern recognition receptors (PRRs), including the canonical Toll-like receptors (TLRs), play a vital role in innate immunity by recognizing and defending against invading pathogens. Specifically, TLR-4 perceives both exogenous and endogenous stimuli, including Microbial/Pathogen/Damage Associated Molecular Patterns (MAMP/PAMP/DAMP) [7,8]. Myeloid Differentiation Factor 88 (MyD88), a TLR adapter protein, mediates signal transduction upon receptor activation. MyD88 recruits tumor necrosis factor receptor-associated factor 6 (TRAF6), interleukin-1 receptor associated Kinase 1 (IRAK1), and IRAK4, and then forms a TLR signaling complex. This subsequently triggers the NF-κB signal cascade to produce various inflammatory cytokines, including IL-1α, IL-1β, tumor necrosis factor (TNF)-α, and IL-6 [9,10,11]. The downstream Phosphatidylinositol 3-kinase (PI3K) targets Protein kinase B (PKB), also called Akt, which is linked to NF-κB activation. This NF-κB activation directs that it has an important impact on PI3K/Akt signal transduction [12]. Therefore, Akt attenuation can suppress iNOS and COX-2, as well as proinflammatory cytokines. Disruption of this pathway could be a promising approach to inhibit the harmful effects of inflammation. 

During inflammation, cells are exposed to recurrent changes in levels of oxidative stress. Hence, oxidative stress and inflammation are closely related, and they drive the central pathophysiological processes of various diseases [13]. Previous studies demonstrated that Nrf2 and Kelch-like ECH-associated protein 1 (Keap1), which are involved in the antioxidant pathway, plays an essential role during inflammatory disease and oxidative stress conditions. Under homeostatic conditions, Nrf2 is associated with its repressor Keap1, which leads to its ubiquitin-dependent degradation in the proteasome [14,15]. Upon activation, a stable Nrf2 translocates into the nucleus where it binds to antioxidant response element (ARE)-containing promoter regions and expresses a large number of phase II genes, such as heme oxygenase-1 (HO-1), NAD (P) H: quinone oxidoreductase-1 (NQO-1), and γ-glutamate-cysteine ligase catalytic subunit (GCLC). Cumulative studies confirmed that induction of these antioxidant cascades not only saves organisms from oxidative damage but also protects against inflammation [16].

Synthetic non-steroidal anti-inflammatory drugs (NSAIDs) and antioxidants are frequently used in treating acute and chronic inflammatory diseases as well as oxidative damage. However, prolonged use of these synthetic small molecule compounds can result in severe side effects, such as gastrointestinal diseases [17]. Hence, it is of great interest to develop novel bioactive peptides. Due to their immunomodulatory properties, small peptides (~1 kDa) have become popular target molecules for therapeutic development. Several natural and synthetic peptides are currently being developed as new therapeutic agents for systemic, lung, and wound infection via activation of pro- and anti-inflammatory responses [18,19]. While it is well known that antimicrobial peptides (AMPs) exert antimicrobial activities primarily via membrane disruption, the underlying mechanism of their anti-inflammatory activity remains unknown. Very few studies attributed the antioxidant and anti-inflammatory activity of AMPs to the upregulation of the Nrf2 pathway and the suppression of the NF-κB pathway, respectively [3,20]. However, the molecular mechanism these peptides use to modulate cellular pathways remains unclear. In this study, we demonstrate that YD1, an AMP consisting of ten amino acids, affects its anti-inflammatory and antioxidant activity by activating the Nrf2/HO-1 pathway, which suppresses the TLR4/MyD88/NF-κB signaling cascade.

## 2. Materials and Methods

### 2.1. Drugs and Chemicals

LPS (from *Escherichia coli* 055:B5, Sigma-Aldrich, St. Louis, MO, USA), dimethylsulfoxide (DMSO), dexamethasone, and tert-butyl hydroperoxide (t-BHP) were obtained from Sigma-Aldrich (St. Louis, MO, USA). Antibiotics (streptomycin, penicillin), Dulbecco’s modified Eagle’s medium (DMEM), and fetal bovine serum (FBS) were obtained from Invitrogen-Gibco (Grand Island, NY, USA). Biolegend (San Diego, CA, USA) provided the mouse TNF-α, IL-1β, and IL-6, using enzyme-linked immunosorbent assay (ELISA) kits. Antibodies against NF-κB, COX-2, iNOS, ERK1/2, p-ERK1/2, JNK, p-JNK, p38, p-p38, β-actin, and Lamin B were procured from Cell Signaling (Boston, MA, USA), Abcam (Cambridge, MA, USA), or Santa Cruz Biotechnology (Dallas, TX, USA). The horseradish peroxidase-conjugated anti-rabbit or anti-mouse IgG was procured from Protein-Tech (Boston, MA, USA). All PCR primers were obtained from Bioneer (Daejeon, Korea). Unless explicitly stated, the remaining chemicals were from Sigma-Aldrich (St. Louis, MO, USA).

### 2.2. YD1 Production and Purification

The production and purification of YD1 from *Bacillus amyloliquefaciens CBSYD1* were carried out as previously described by Rahman [2,3].

### 2.3. Cell Viability Measurement

RAW 264.7 cells were inoculated at a 2 × 10^5^ cells/mL concentration in a 96-well plate and incubated at 37 °C for 24 h. Then, the predetermined concentrations of YD1 (2.5–80 μg/mL) or the vehicle were added to the cells for 20 h. The culture medium was removed, and 100 μL of 10% MTT (3-(4,5-dimethylthiazol-2-yl)-2,5-diphenyltetrazolium bromide) solution was added to the cells and incubated for 1 h at 37 °C. Colored formazan crystals were dissolved with a 100% DMSO solution, and the optical density (OD) was measured at 590 nm using a microplate reader (PerkinElmer, Waltham, MA, USA) [21].

### 2.4. Measurement of NO, TNF-α, IL-1β, IL-6, IL-10, and PGE_2_

RAW 264.7 cells were treated with predetermined concentrations of YD1 (2.5, 5, and 10 μg/mL) with or without LPS (1 μg/mL) for 24 h. Then, cell culture supernatants were collected, and NO was determined using the Griess reagent [22]. ELISA was utilized to determine the levels of TNF-α, IL-1β, IL-6, and PGE2, per the manufacturer’s protocol.

### 2.5. Intracellular ROS Generation Determination

The production of reactive oxygen species (ROS) in LPS-treated RAW 264.7 cells was measured spectrofluorometrically using the DCFH-DA method [23]. The YD1 (2.5–10 µg/mL) pretreated cells were stimulated with LPS (1 µg/mL) for 12 h. Then, phosphate-buffered saline (PBS) was used to wash the cells twice, and they were then treated with 25 µM DCF-DA at 37 °C for 30 min, after which the fluorescence intensity was measured using a fluorescence microplate reader. The wavelength values for excitation and emission were recorded at 485 nm and 528 nm, respectively (PerkinElmer, Waltham, MA, USA). The cells were also analyzed under confocal microscopy (Carl Zeiss, Jena, Germany).

### 2.6. NF-κB Reporter Assay

RAW 264.7 cells were inoculated in 12-well plates and transfected with a mixture containing the reporter gene pNF-κB-luc, the reporter constructs pRL SV40, and the Transfection Reagent for 24 h. After incubation with YD1 (2.5, 5, and 10 μg/mL) for 30 min, transfected cells were stimulated with LPS (1 µg/mL). The Luciferase assay was performed using the Dual-Luciferase Reporter Assay System (Promega, Madison, WI, USA) as described by Alam [21]. Immunocytochemistry and confocal image analysis of the NF-κB translocation were carried out as described by Ik-Soo Lee [24].

### 2.7. ARE Promoter Activity

RAW 264.7 cells were inoculated at a 1 × 10^5^ cells/mL concentration in a 24-well plate and incubated at 37 °C for 24 h. Using the ViaFect Transfection Reagent (Promega, Madison, WI, USA), pGL4.74 (hRluc/TK vector) and pGL4.37 (luc2P/ARE/Hygro vector) plasmids were transfected into the cells. Then, predetermined concentrations of YD1 (2.5, 5, and 10 µg/mL) were added, and the cells were incubated for 1, 3, 6, 12, and 18 h. To determine the ARE-driven promoter activity, a Dual-Luciferase Reporter Assay System (Promega, Madison, WI, USA) was used.

### 2.8. Transfection of Small Interfering RNA (siRNA)

RAW 264.7 cells (1 × 10^5^ cells/mL) were inoculated at a 1 × 10^5^ cells/mL concentration in a 6-well plate for 24 h. Using Lipofectamine RNAiMax (Invitrogen, Carlsbad, CA, USA), the cells were transfected with 10–50 nM siRNA as per manufacturer’s protocol. The si-Nrf2 RNAs and si-Control were obtained from Santa Cruz Biotechnology (catalog number: SC-37049, Santa Cruz, CA, USA).

### 2.9. Animals

Mice at 8 weeks weighing 25–30 g, obtained from the Institute of Cancer Research (ICR) Samtako (Osan-si, Gyenonggi-do, South Korea), were housed at a controlled temperature (23 ± 1 °C), humidity (55 ± 5%), and 12 h light/dark cycle. They were fed with standard water ad libitum and rodent chow for 1 week. The animal study was approved by the Institutional Animal Care and Use Committee (IACUC) at Kyungpook National University (KNU) under the code KNU-2017-0010. Twenty-four mice were randomly divided into four groups (6 animals/group). Group 1 was set as the control and received dH_2_O. Group 2 was considered as the carrageenan (CA) control group. In contrast, group 3 and 4 received indomethacin (10 mg/kg, p.o.) and YD1 (10 mg/kg, p.o.), respectively. Group 4 was treated with YD1 (10 mg/kg/day, p.o.) for 4 days, whereas indomethacin, an anti-inflammatory drug (10 mg/kg, positive control, p.o.), was administered 1 h before the CA insult. To induce acute-inflammation, CA (1% in saline solution, 60 μL/mice) was administered subcutaneously in their right hind paw. Paw edema volume was measured before and every hour for the duration of 4 hours after the CA insult using a plethysmometer (UGO BASILE; Comerio, VA, Italy). Subsequently, the animals were euthanized, and the skin of the right hind legs was dissected and immediately stored at −80 °C for further analysis.

### 2.10. RT-PCR, Protein Extraction, and Western Blotting Analysis

Reverse transcription-polymerase chain reaction (RT-PCR), protein extraction, and western blotting were conducted as previously described by Rahman et al. [2]. All primer sequences and antibodies information are provided in the Appendix A.

### 2.11. Statistics

Student’s *t*-test was used to determine the statistical significance using SPSS statistical software. The statistically significant results were defined as *p* < 0.05, where the differences in the YD1-treated and non-treated samples were analyzed by a post hoc test using the Tukey HSD test.

## 3. Results

### 3.1. YD1 Inhibits Inflammatory Responses

To assess the anti-inflammatory properties of YD1, NO and PGE_2_ production was measured in cultured murine macrophages (RAW 264.7) treated with LPS to induce inflammation, then subsequently treated with a non-toxic concentration of YD1 (Appendix A). As shown in Figure 1A,B, LPS insult significantly induced NO and PGE_2_ compared with untreated cells. Treatment with YD1 significantly decreased NO and PGE_2_ production in a concentration-dependent fashion (Figure 1A,B, respectively). Interestingly, pretreatment with YD1 or L-NIL, a selective inhibitor of iNOS used as a positive control, suppressed LPS-induced NO generation by 6.2- and 5.8-fold, respectively (2.5–10 μg/mL), compared with just LPS-treated cells. Likewise, pretreatment with YD1 or NS-398, a specific COX-2 inhibitor, suppressed LPS-induced PGE_2_ production by 8.5- and 8.8-fold, respectively, (2.5–10 μg/mL), compared to just LPS-treated cells.

To examine whether the production of NO and PGE_2_ is modulated by the transcription or translation of COX-2 and iNOS, both expression and protein levels were measured using RT-PCR and immunoblotting assays. LPS treatment significantly increased the mRNA and protein levels of both COX-2 and iNOS, whereas YD1 pretreatment decreased the mRNA and protein levels of COX-2 and iNOS in a concentration-dependent manner in LPS-treated cells (Figure 1C,D). These observations indicate that YD1 inhibits the LPS-stimulated inflammatory response by suppressing the expression of COX-2 and iNOS in RAW 264.7 cells.

Likewise, to investigate the effects of YD1 on proinflammatory cytokine production, we performed RT-PCR and immunoblotting assay. LPS insult significantly up-regulated both mRNA expression and the protein abundance of TNF-α, IL-1β, and IL-6 (Figure 1E,F). Treatment with YD1 dramatically attenuated the increase in proinflammatory cytokines induced in the LPS-treated RAW 264.7 cells (Figure 1E,F).

### 3.2. YD1 Inhibits LPS-Induced TLR4/MyD88/NF-κB Signaling

TLR4 signaling plays a crucial role in the immune response. LPS insult can activate TLR4 in macrophages, thereby modulating the NF-κB signaling pathway [25]. To examine the effects of YD1 on the TLR4-mediated signaling pathway, TLR4, Myd88, IRAK4, and TRAF6 protein abundance was analyzed via immunoblotting. We found that LPS insult significantly increased the protein abundance of all proteins assayed compared to untreated RAW 264.7 cells. In contrast, when the cells were pretreated with YD1, the effects of LPS on the proteins associated with the TLR4 signaling pathway were significantly attenuated in a concentration-dependent manner (Figure 2A). These results indicate that YD1 could regulate the TLR4/MyD88-dependent pathway via modulation of the downstream proteins, such as IRAK4 and TRAF6.

Next, we investigated whether YD1 could prevent the phosphorylation of IκBα. We found that YD1 pretreatment notably reduced the LPS-induced IκBα phosphorylation in a concentration-dependent manner (Figure 2B). Thus far, our results suggest that the immunomodulatory effects of small peptides/glycoproteins from natural products are closely associated with NF-κB signaling. We further examined whether YD1 hinders NF-κB activation in LPS-stimulated RAW 264.7 cells. We found that LPS treatment augmented the expression of NF-κB while YD1 treatment significantly abrogated the transcriptional effects of LPS on NF-κB (Figure 2C). Next, we sought to discover if LPS treatment had any effect on the nuclear translocation of NF-κB by utilizing a reporter assay. Our results demonstrate that the nuclear translocation of NF-κB is significantly suppressed by YD1 in a concentration-dependent manner in LPS-induced cells (Figure 2D). These results provide further evidence that YD1 abrogates LPS-induced NF-κB activation.

Through IκB degradation, Akt promotes NF-κB activation [26]. Consequently, as shown in Figure 2E, treatment with LPS increased Akt phosphorylation significantly. However, significant inhibition was observed in LPS-induced Akt phosphorylation by YD1. This result showed the involvement of the Akt signaling molecule in the inflammatory response.

### 3.3. Nrf2-ARE Signaling and Its Downstream Phase II Antioxidant Enzymes

At resting conditions, cytosolic Nrf2 is inactivated upon binding to its inhibitor Keap1. Activated Nrf2 translocates to the nucleus after dissociating from Keap1 and binds directly to AREs in target promoters, inducing the expression of several antioxidant genes, including HO-1 and NQO-1 [21,27]. Previous studies demonstrated the association between Nrf2 and NF-κB signaling [24,28]. As our results thus far demonstrated that YD1 suppresses NF-κB activation, we next sought to determine if YD1 also affects Nrf2 signaling. We monitored ARE-promotor activity upon treatment with YD1 using a Dual-Luciferase reporter assay. Treatment with YD1 (2.5–10 μg/mL) which resulted in a notable increase in ARE-luciferase activity over time with a peak value at 3 h (4 ± 0.08-fold) in RAW 264.7 cells (Figure 3A). We next examined the nuclear translocation of Nrf2 when treated with YD1 utilizing immunoblotting analysis. Within an hour, YD1-treated cells showed translocation of NRF2 into the nucleus (Figure 3B). Furthermore, in the investigation described in Figure 3C, YD1 increased NQO1, HO-1, and antioxidant enzymes. These findings suggest that YD1 activates the Nrf2-ARE signaling cascade in addition to suppressing the NF-κB pathway.

Transfection of the cells with Nrf2 siRNA before YD1 treatment significantly reduced Nrf2 expression, as expected, which eliminated the expression of HO-1 (Figure 3D). These results confirm that the YD1-induced expression of HO-1 is mediated via the Nrf2 pathway.

The spectrofluorometric analysis showed that LPS exposure resulted in the accumulation of intracellular ROS levels and that YD1 treatment abrogated the ROS accumulation (Figure 3E). This antioxidant effect was comparable to the effects of Gallic acid, an antioxidant with well-known potent ROS-reducing potential (Figure 3F). This reduction in ROS levels was also confirmed via confocal microscopic analysis (Figure 3F). To understand how YD1 impacts the Nrf2 pathway in antioxidant activity, a complementation study was conducted. This study revealed that YD1-induced ROS scavenging was only partially reduced in the presence of brusatol, a pharmacologically specific Nrf2 inhibitor (Figure 3G), indicating that the antioxidant effects of YD1 are only partially enacted via the Nrf2 pathway.

### 3.4. YD1 Suppresses NF-κB Activation via Nrf2/HO-1-Mediated Pathway

To determine if YD1 suppresses NF-κB activation through inducing HO-1 expression, we examined the effects of an HO-1 inhibitor, SnPP, on p-IκBα expression, the translocation of NF-κB, the generation of NO and PGE_2_, as well as the expression of macrophage biomarkers (e.g., IL-1β, IL-6, and TNF-α). We found that SnPP (10 μg/mL) abolished the YD1 inhibitory effect on LPS-induced p-IκBα expression and nuclear translocation of NF-κB (Figure 4A–C). In addition, we found that SnPP halted the suppressive effects of YD1 on NO, PGE_2_, TNF-α, IL-1β, and IL-6 (Figure 4D,E). Taken together, our data suggest that the anti-inflammatory action of YD1 is accomplished by suppression of the NF-κB inflammatory pathway via up-regulation of HO-1 in RAW 264.7 cells.

### 3.5. YD1 Effects on Mouse Paw Edema

We next studied the effects of YD1 in a carrageenan (CA) inflammatory mouse model. We found that YD1 treatment (10 mg/kg/day, p.o.) reduced CA-induced paw edema volume by 1.9-fold, which is comparable to the effects of Indometacin, a popular NSAID (Figure 5A,B). Additionally, we found that YD1 treatment significantly up-regulated the levels of HO-1, NOQ1, and Nrf-2 proteins while down-regulating the expression of iNOS, COX-2, p-IκBα, and NF-κB compared to the non-YD1-treated groups (Figure 5C,D). Our results indicate that YD1 alleviates CA-induced inflammation in mice, enacted through the activation of the Nrf2/HO-1 signaling cascade, which suppresses inflammatory NF-κB signaling.

## 4. Discussion

Peptide drugs have attracted the attention of researchers and biomedical industries because of their wide range of antibiotic and immunomodulatory activities. In our previous study, we characterized an antimicrobial and antioxidant peptide, YD1, from the *Bacillus amyloliquefaciens CBSYD1* strain isolated from the Korean traditional fermented food, kimchi. In this work, we investigated the mechanism of YD1 in anti-inflammatory reactions. Specifically, we studied the impact of YD1 on Nrf2 and NF-κB signaling in both an in vitro and in vivo model.

Activated macrophages play a crucial role in the inflammatory processes [29] and cause mass production of inflammatory mediators, for example, NO and PGE2, and proinflammatory cytokines such as TNF-α, IL-1β, and IL-6, leading to the progression of various inflammatory disorders [24,28]. A few small molecular weight plants and microbial peptides showed anti-inflammatory potential by affecting the production of PGE_2_ and down-regulating genes related to inflammation in RAW 264.7 cells [30,31,32]. We discovered that YD1 remarkably impeded the production of NO and PGE_2_ by suppressing the expression of iNOS and COX-2. The inhibition of iNOS and COX-2 can exert significant immunosuppressive effects [33,34]. Proinflammatory cytokines play a primary role in the inflammatory response; specifically, cytokine IL-6 is found to interact with many cellular targets, is linked to diverse immunological reactions [35,36], and is involved in initiating and regulating the inflammatory process [37]. Accordingly, the suppression of these proinflammatory cytokines could be a potential target for anti-inflammatory drug development. Our results show that YD1 treatment significantly suppressed the mRNA expression of several proinflammatory cytokines, including IL-6, demonstrating its anti-inflammatory capabilities.

Many cells, such as neutrophils, endothelial cells, macrophages, and dendritic cells, possess TLRs [38]. Among the various TLRs, LPS is only recognized by TLR4. The TLR4-mediated pathway plays an important role in the induction of the inflammation response. The LPS-stimulated TLR4 facilitates the recruitment of the TLR Signaling Complex, which leads to the activation of NF-κB and subsequent expression of proinflammatory cytokines. We found that YD1 suppresses both the activation of TLR4 and the MyD88-mediated NF-κB signaling pathway through IκBα phosphorylation as well as degradation in a concentration-dependent manner in Figure 2. These results suggest that the inflammation induced by LPS can be suppressed by YD1 via modulation of the TLR4/MyD88/NF-κB signaling pathway. Further study would be interesting to look at the effect of YD1 on IKK activation, determined by immunoblotting with a phospho-IKK antibody. Further, the effect of YD1 on TLR4 activation of p38a and c-Jun N-terminal kinases (JNKs), as they belong to the mitogen-activated protein kinase (MAPK) family and are responsive to stress stimuli, can be determined to establish the specificity of YD1, and its effects on NF-kB and MAP kinase activation by TNF. In addition, the study of YD1 effects on TLR4 activation of activator protein 1 (AP-1), a transcription factor that regulates the expression of genes in response to a variety of stimuli, including cytokines, using the appropriate reporter assay will provide a firm picture by which NF-kB signaling is controlled by YD1.

A recent study revealed that certain bioactive proteins attenuate wound-healing by activating the Nrf2/HO-1 signaling cascade [39]. HO-1 is one of the critical downstream genes of the Nrf2 signaling cascade that plays a key role in cytoprotection against oxidative damage and inflammation by inhibiting the activation of NF-κB and the subsequent production of proinflammatory cytokines [40,41]. We revealed that YD1 activates Nrf2 signaling, which suppresses inflammation through the production of the antioxidant enzymes HO-1 and NQO1, which suppress the phosphorylation of IκBα and subsequent NF-κB activation. To this end, we demonstrated that YD1 has similar anti-inflammatory effects and acts via the same Nrf2 pathway in an in vivo model using carrageenan (CA)-induced paw edema in mice. These findings confirmed the contribution of Nrf2-induced HO-1 expression to the anti-inflammatory effects of YD1. However, it will be interesting to elucidate the complete structural information of the YD1 peptide and evaluate the YD1 analogs for biological activities.

The present study demonstrated initial evidence that YD1, isolated from the *Bacillus amyloliquefaciens CBSYD1,* exerted anti-inflammatory effects in LPS-induced macrophages via the inhibition of inflammatory mediators and their associated genes, as well as a reduction in proinflammatory cytokines, and showed a crosstalk mechanism of regulation of inflammation processes involving HO-1, and the balance between NF-κB and Nrf2 activities (Figure 6). Our study strongly demonstrates YD1 as a potential novel anti-inflammatory agent, whereas further systemic analyses are needed in LPS-challenged animals to determine the in vivo efficacy and pharmacokinetics of YD1. Future studies could help determine if YD1 could be a potential therapeutic for preventing and treating inflammation-related disorders and oxidative stress.

## Figures and Tables

**Figure 1 ijms-22-05161-f001:**
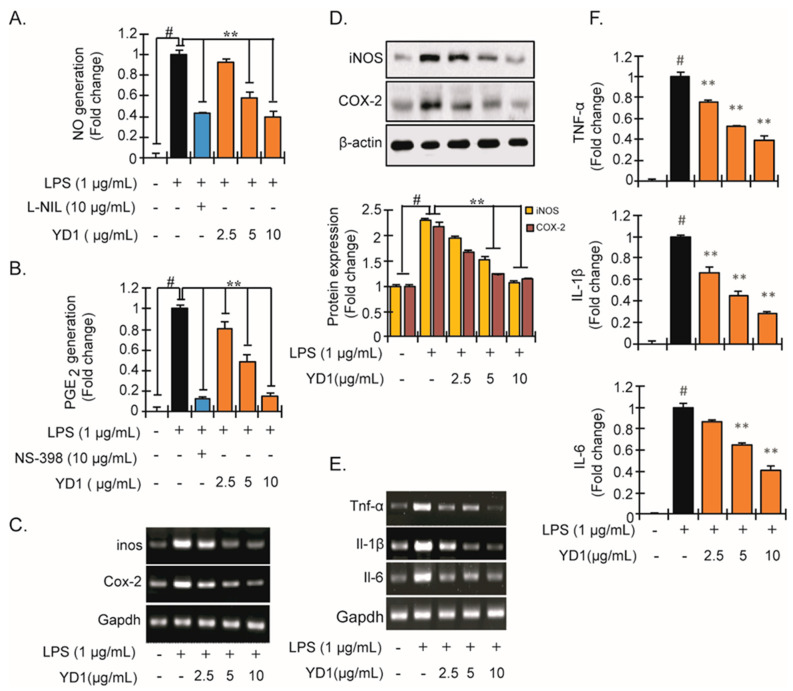
Inhibitory effect of YD1 on various inflammatory mediators in LPS-stimulated RAW 264.7 cells. Cells (5 × 105 cells/mL) were treated with various concentrations (f.c. 2.5, 5, and 10 μg/mL) of YD1 for 1 h and then treated with LPS (f.c. 1 µg/mL) for 24 h. NO generation was measured by Griss reagent (**A**); PGE2 productions was confirmed by ELISA (**B**); mRNA levels of iNOS and COX-2 were confirmed by RT-PCR (**C**); Protein expression of iNOS and COX-2 was identified by immunoblotting (**D**); mRNA expression of TNF-α, IL-1β, and IL-6 was confirmed by RT-PCR (**E**), and their corresponding productions were identified by ELISA assay (**F**). Densitometric analysis was carried out to quantify the band intensity by β-actin normalization. Results are expressed as the mean ± SD of three separate experiments. ^#^
*p* < 0.05 statistically different from untreated cells; ** *p* < 0.05 statistically different from LPS alone.

**Figure 2 ijms-22-05161-f002:**
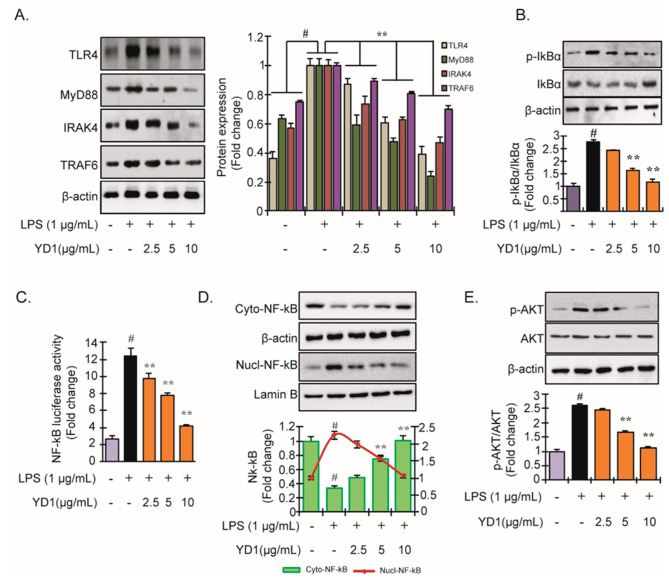
Inhibitory effect of YD1 on TLR4/MyD88/NF-κB signaling in LPS-stimulated RAW 264.7 cells. Cells (5 × 10^5^ cells/mL) were treated with various concentrations (f.c. 2.5, 5, and 10 μg/mL) of YD1 for 1 h and then treated with LPS (f.c. 1 μg/mL) for 20 h for the protein levels of TLR4, MyD88, IRAK4, and TRAP6 (**A**); and for 1 h for p-IκBα and IκBα (**B**) in YD1-treated RAW 264.7 cells, which were analyzed by immunoblotting. NF-κB reporter gene assay was carried out according to the procedure described in materials and methods (**C**). Cells were pretreated with YD1 (2.5, 5, and 10 μg/mL) for 1 h and subsequently co-treated with LPS (1 μg/mL) for an additional 2 h, and the protein level of NF-κB in cytosol (upper panel) and nucleus (lower panel) was determined by immunoblotting (**D**). RAW 264.7 cells were pretreated with YD1 (2.5, 5, and 10 μg/mL) for 1 h and then stimulated with LPS (1 μg/mL) for 1 h, and phosphorylation levels of Akt were analyzed using immunoblotting (**E**). Densitometric analysis was carried out to quantify the band intensity by β-actin normalization. Results are expressed as the mean ± SD of three separate experiments. ^#^
*p* < 0.05 statistically different from untreated cells; ** *p* < 0.05 statistically different from LPS alone.

**Figure 3 ijms-22-05161-f003:**
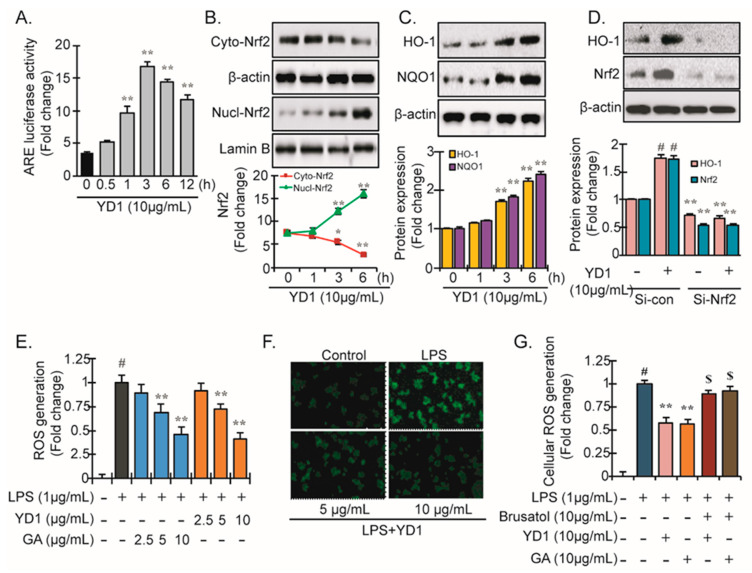
YD1 activates Nrf2-ARE signaling. RAW 264.7 cells were treated with YD1 (10 μg/mL) for the indicated time durations, and ARE luciferase activity was measured by a Dual Luciferase reporter assay as described in Materials and Methods (**A**). The levels of Nrf2 protein in cytosolic and nuclear fractions were analyzed by Western blot analysis (**B**). RAW 264.7 cells were treated with YD1 (10 μg/mL), and protein levels of HO-1 (upper panel) and NQO1 (lower panel) were analyzed by Western blotting (**C**). Densitometric analysis was carried out to quantify the band intensity by β-actin normalization. Data (mean ± SD) were representative of at least three independent experiments and expressed as the fold-induction relative to untreated cells (at time zero); ** *p* < 0.05 versus untreated control. RAW 264.7 cells were transiently transfected with Nrf2 siRNA for 24 h followed by YD1 treatment and incubated for 6 h. Nrf2 protein levels were measured by Western blotting analysis (**D**). Cells were pretreated by YD1 (2.5, 5, and 10 μg/mL) for 12 h, and subsequently stimulated by LPS for 24 h. The production of intracellular ROS was assayed using the fluorescent probe DCFH-DA (**E**). Immunofluorescent confocal microscopy showed the level of ROS (green fluorescence) in RAW 264.7 cells (**F**). Cells were treated with brusatol (a pharmacologically specific Nrf2 inhibitor) for 30 min, followed by YD1 and (10 μg/mL) and gallic acid (GA) (10 μg/mL) for 12 h and then subjected to LPS insult for 24 h, and the cellular ROS generation was measured (**G**). ^#^
*p* < 0.05 compared to the untreated cells; ** *p* < 0.05 compared to UVB-treated cells. ^$^
*p* < 0.05 compared with YD1 and gallic acid-treated cells.

**Figure 4 ijms-22-05161-f004:**
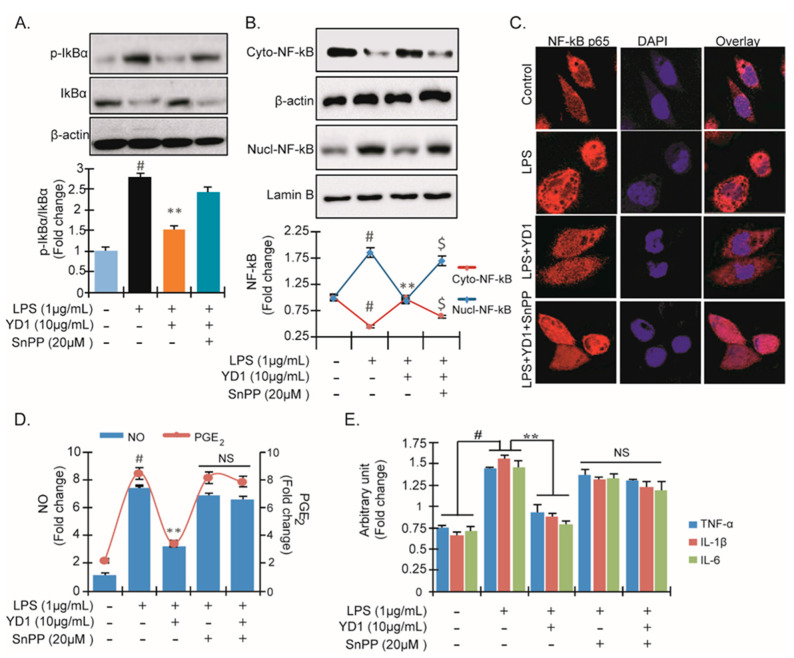
Anti-inflammatory effect of YD1 through the contribution of HO-1 induction. RAW 264.7 cells were exposed to 10 μg/mL YD1 for 1 h in the presence or absence of HO-1 inhibitor SnPP (20 mM), and were subsequently treated with LPS for an additional 1 h for p-IκBα and IκBα protein (**A**) and 2 h for NF-κB protein level in cytosol (upper panel) and in nucleus (lower panel), which were analyzed by Western blot analysis (**B**). Densitometric analysis was carried out to quantify the band intensity by β-actin normalization. Immunofluorescent confocal microscopy showed the level and location of NF-κB (red fluorescence) in RAW 264.7 cells. Hoechest staining (blue fluorescence) represents the nuclei (**C**). Degree of NO production (**D**) and TNFα, IL-1β, and IL-6 generation (**E**) were expressed as mean ± SD. ^#^
*p* < 0.05 versus untreated control, ** *p* < 0.05 versus LPS alone-treated control; ^$^: nonsignificant compared to untreated control; NS: nonsignificant.

**Figure 5 ijms-22-05161-f005:**
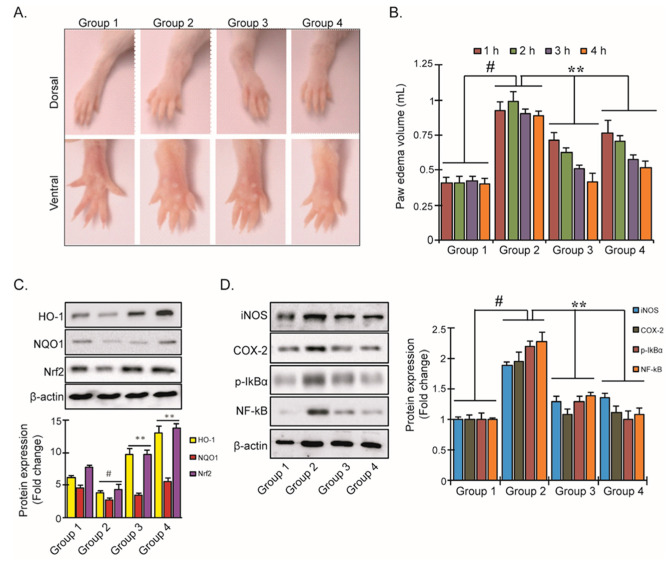
Inhibition of carrageenan (CA)-induced paw edema by YD1. All animals in each group received λ-Carrageenan (5 mL/kg body weight (i.p)) except group 1, which served as the regular saline control (5 mL/kg body weight (i.p)). Dorsal and ventral view of mice paw after 4 h of CA-induction (**A**). The degree of paw volumes was measured 0–4 h after CA injection, as described in Materials and Methods (**B**). Protein expression of HO-1, NQO-1, and Nrf2 (**C**) as well as iNOS, COX-2, p-IκB, and NF-κB in CA-induced mice (**D**). Densitometric analysis was carried out to quantify the band intensity by β-actin normalization. Results represent the mean ± S.D. of six animals. ^#^
*p* < 0.05 significant compared with vehicle-treated control; ** *p* < 0.05 significant compared with CA insult group.

**Figure 6 ijms-22-05161-f006:**
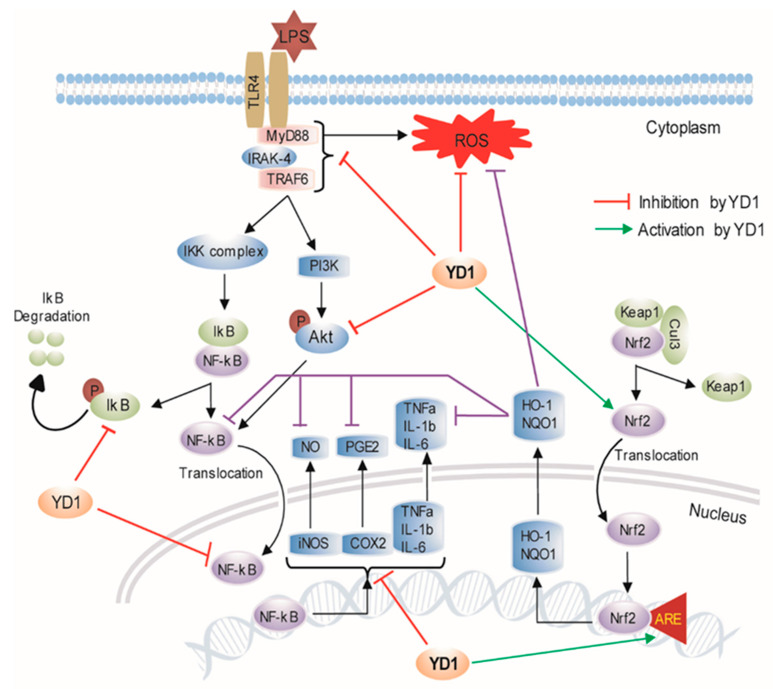
A hypothetical model is illustrating the potential role of YD1 for anti-inflammatory activity.

## Data Availability

The data that support the findings of this study are openly available in GenBank (ncbi.nlm.nih.gov/Genbank) under accession no. KY062987.

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
