# Peer review of "Activation of Nrf2/HO-1 by Peptide YD1 Attenuates Inflammatory Symptoms through Suppression of TLR4/MYyD88/NF-κB Signaling Cascade"

_ijms, 2021, doi:10.3390/ijms22105161_

Round 1
Reviewer 1 Report
In this study, authors studied the immunomodulatory effects of a novel antimicrobial peptide, YD1, isolated from Kimchi, in both in vitro and in vivo models. Authors found that YD1 exerts anti-inflammatory effects via up-regulation of the Nrf2 pathway, resulting in the production of HO-1, which suppresses activation of the NF-κB pathway, including the subsequent proinflammatory cytokines IL-1β, IL-6, and TNF-α. They also found that YD1 suppressed nitric oxide (NO) and prostaglandin E2 (PGE2) production by down-regulating the expression of the upstream genes, iNOS, and COX-2.
I do appreciate the ideas of this work, results and discussions. I have few questions/suggestions that I would like to ask authors.
- In the introduction, I think it would be good if authors could mention about why authors have chosen to study anti-inflammatory activity of YD1 using murine macrophage RAW 264.7 cells. What are the benefits of using murine macrophage instead of human macrophages?
- For cell viability measurement, authors mentioned that RAW 264.7 cells were inoculated at 1 × 105 cells/mL concentration in a 96-well plate but in the supplementary figure 1 authors mentioned that RAW 264.7 cells were seeded at a density of 2 × 104 cells per well (96-well plate). Could authors clarify this?
- In Figure 1, I think it would be good if authors could re-write TNF-α, IL-1β, IL-6 and iNOS, COX-2 and GAPDH in consistent way.
- In Supplementary figure 1, please rewrite this “at a density of 2 × 104 cells per well”
- What would it be the future study of biological activities of this peptide? From my understanding, antioxidant, antimicrobial and anti-inflammatory activities are closely related but there are also related to other biological activities as well.
Author Response
Reviewer -1:
In this study, authors studied the immunomodulatory effects of a novel antimicrobial peptide, YD1, isolated from Kimchi, in both in vitro and in vivo models. Authors found that YD1 exerts anti-inflammatory effects via up-regulation of the Nrf2 pathway, resulting in the production of HO-1, which suppresses activation of the NF-κB pathway, including the subsequent proinflammatory cytokines IL-1β, IL-6, and TNF-α. They also found that YD1 suppressed nitric oxide (NO) and prostaglandin E2 (PGE2) production by down-regulating the expression of the upstream genes, iNOS, and COX-2.
I do appreciate the ideas of this work, results and discussions. I have few questions/suggestions that I would like to ask authors.
- In the introduction, I think it would be good if authors could mention about why authors have chosen to study anti-inflammatory activity of YD1 using murine macrophage RAW 264.7 cells. What are the benefits of using murine macrophage instead of human macrophages?
Ans: We appreciate for valuable comments; we have revised the introduction as suggested (line 45-51)
- For cell viability measurement, authors mentioned that RAW 264.7 cells were inoculated at 1 × 105 cells/mL concentration in a 96-well plate but in the supplementary figure 1 authors mentioned that RAW 264.7 cells were seeded at a density of 2 × 104 cells per well (96-well plate). Could authors clarify this?
Ans: We apologise. Thanks for pointing this. This must be a typo. We have corrected it. It should be 2 × 105
- In Figure 1, I think it would be good if authors could re-write TNF-α, IL-1β, IL-6 and iNOS, COX-2 and GAPDH in consistent way.
Ans: Thanks for commenting on this. We have revised and all authors agree that no further re-arrangement/re-writing is requiring for fig-1.
- In Supplementary figure 1, please rewrite this “at a density of 2 × 104 cells per well”
Ans: Thanks. We have corrected
- What would it be the future study of biological activities of this peptide? From my understanding, antioxidant, antimicrobial and anti-inflammatory activities are closely related but there are also related to other biological activities as well.
Ans: Thanks for commenting on this. We have revised the discussion as suggested (line 357-359)
Reviewer 2 Report
The authors showed investigated the immunomodulatory effects of a short peptide, YD1, its anti-inflammatory effects via up-regulation of the Nrf2 pathway, resulting in the production of HO-1, which suppresses activation of the NF-κB pathway with cytokines IL-1β, IL-6, and TNF-α. YD1 suppressed NO and PGE2 production by down-regulating the expression of the upstream genes, iNOS, and COX-2.
The experiments are well-performed, and the manuscript is well written. The authors already reported in the previous paper (ref. 30) that YD1 treatment on RAW 264.7 cells increased the transcriptional and translational activities of Nrf-2 through the enhanced levels of HO-, and YD1 contains a strong antioxidant activity by decreasing nitric oxide (NO) and reactive oxygen species (ROS) in RAW 264.7 cells. The authors should mention the relationship between the previous and present work in the discussion section.
line 284, “suggests” should be “suggest”.
Author Response
Reviewer -2:
Comments and Suggestions for Authors
The authors showed investigated the immunomodulatory effects of a short peptide, YD1, its anti-inflammatory effects via up-regulation of the Nrf2 pathway, resulting in the production of HO-1, which suppresses activation of the NF-κB pathway with cytokines IL-1β, IL-6, and TNF-α. YD1 suppressed NO and PGE2 production by down-regulating the expression of the upstream genes, iNOS, and COX-2.
The experiments are well-performed, and the manuscript is well written. The authors already reported in the previous paper (ref. 30) that YD1 treatment on RAW 264.7 cells increased the transcriptional and translational activities of Nrf-2 through the enhanced levels of HO-, and YD1 contains a strong antioxidant activity by decreasing nitric oxide (NO) and reactive oxygen species (ROS) in RAW 264.7 cells. The authors should mention the relationship between the previous and present work in the discussion section.
Ans: We appreciate for valuable comments. In this study, we establish the cross-talk mechanism of regulation of inflammation processes involving HO-1, and the balance between NF-κB and Nrf2 activities, we have mentioned (line: 360-369). Therefore, all authors agree it is important to reassure some antioxidant activity.
Line 284, “suggests” should be “suggest.”
Ans: Thanks. We have corrected